# The Role of Extracellular Signal-Regulated Kinase Pathways in Different Models of Cardiac Hypertrophy in Rats

**DOI:** 10.3390/biomedicines11092337

**Published:** 2023-08-22

**Authors:** Gassan Moady, Offir Ertracht, Efrat Shuster-Biton, Elias Daud, Shaul Atar

**Affiliations:** 1The Cardiology Department, Galilee Medical Center, Nahariya 2210001, Israel; eliasdaud57@yahoo.de (E.D.); shaula@gmc.gov.il (S.A.); 2The Azrieli Faculty of Medicine, Bar-Ilan University, Safed 1311502, Israel; efratshus@gmail.com; 3The Cardiovascular Research Laboratory, Medical Research Institute, Galilee Medical Center, Nahariya 2210001, Israel; offire@gmc.gov.il

**Keywords:** rat model, hypertrophy, remodeling, eccentric, concentric

## Abstract

Cardiac hypertrophy develops following different triggers of pressure or volume overload. In several previous studies, different hypertrophy types were demonstrated following alterations in extracellular signal-regulated kinase (ERK) pathway activation. In the current study, we studied two types of cardiac hypertrophy models in rats: eccentric and concentric hypertrophy. For the eccentric hypertrophy model, iron deficiency anemia caused by a low-iron diet was implemented, while surgical aortic constriction was used to induce aortic stenosis (AS) and concentric cardiac hypertrophy. The hearts were evaluated using echocardiography, histological sections, and scanning electron microscopy. The expression of ERK1/2 was analyzed using Western blot. During the study period, anemic rats developed eccentric hypertrophy characterized by an enlarged left ventricle (LV) cavity cross-sectional area (CSA) (59.9 ± 5.1 mm^2^ vs. 47 ± 8.1 mm^2^, *p* = 0.002), thinner septum (2.1 ± 0.3 mm vs. 2.5 ± 0.2 mm, *p* < 0.05), and reduced left ventricular ejection fraction (LVEF) (52.6% + 4.7 vs. 60.3% + 2.8, *p* < 0.05). Rats with AS developed concentric hypertrophy with a thicker septum (2.8 ± 0.6 vs. 2.4 ± 0.1 *p* < 0.05), increased LV muscle cross-sectional area (79.5 ± 9.33 mm^2^ vs. 57.9 ± 5.0 mm^2^, *p* < 0.001), and increased LVEF (70.3% + 2.8 vs. 60.0% + 2.1, *p* < 0.05). ERK1/2 expression decreased in the anemic rats and increased in the rats with AS. Nevertheless, the p-ERK and the p-MEK did not change significantly in all the examined models. We concluded that ERK1/2 expression was altered by the type of hypertrophy and the change in LVEF.

## 1. Introduction

Cardiac hypertrophy develops following physiological and pathological triggers and is mediated by several structural and molecular alterations in tissue morphology, volume, and geometry [1,2,3]. Physiological hypertrophy is mainly seen in an athlete’s heart following regular exercise or during pregnancy and is usually reversible, whereas pathological hypertrophy develops following chronic pressure or volume overload conditions [3,4]. In general, chronic pressure overload conditions increase ventricular wall thickness without chamber enlargement, leading to concentric hypertrophy. Chronic volume overload promotes chamber dilatation with no increase or even a decrease in left ventricular (LV) wall thickness to form eccentric hypertrophy [5]. Genetic mutations can also cause concentric hypertrophy (hypertrophic cardiomyopathy) and eccentric hypertrophy (primary dilated cardiomyopathy) [6]. The mitogen-activated protein kinases (*MAPKs*) are a network of protein cascades that play a central role in the mediation of cardiac hypertrophy [7,8]. Four main subfamilies of *MAPKs* are recognized in mammals: c-Jun N-terminal kinases (*JNKs*), p38 mitogen-activated protein kinases (p38s), extracellular signal-regulated kinases (ERK) 1/2, and ERK5. All of them are characterized by phosphorylation-dependent activation [9,10,11,12,13]. The ERK 1/2 cascade is initiated at the cell membrane following various stimuli including growth factors, cytokines, G-protein coupling, and oncogenes [14,15]. The balance between the different forms of cardiac hypertrophies and the activation state of ERK1/2 signaling pathway was demonstrated using transgenic mice models [16,17,18]. In these models, the acute inhibition of ERK1/2 signaling resulted in cardiomyocytes lengthening mimicking eccentric hypertrophy, while constitutive ERK1/2 signaling resulted in increased myocyte thickness consistent with concentric hypertrophy. Nevertheless, the relationship between ERK 1/2 signaling and cardiac hypertrophy types is more complicated and depends on the duration of the stress, the hemodynamic effects (preload and afterload), and cardiac function (preserved vs. reduced contractility). In the current study, we examined the relationship between cardiac hypertrophy and ERK 1/2 signaling in a reverse way by inducing cardiac hypertrophy in rats. For this purpose, we used two models to induce concentric and eccentric cardiac hypertrophy in male Sprague-Dawley rats. For eccentric hypertrophy, a low-iron diet was implemented to induce iron deficiency anemia, whereas ligation of the aorta was performed to induce AS and a concentric hypertrophy model. We hypothesized that the induction of cardiac hypertrophy would alter ERK protein expression depending on the type of hypertrophy and the change in LVEF.

## 2. Materials and Methods

### 2.1. Animal Models and Study Design

All animal experiments were conducted according to the institutional animal ethical committee guidelines, which conform to the *Guide for the Care and Use of Laboratory Animals* published by the US National Institutes of Health (Eighth edition 2011, ethical numbers: 10-02-2012 and 40-10-2013). We used 250–300 g. (6–8 weeks old) male Sprague Dawley rats (Envigo Ltd., Jerusalem, Israel), which were maintained at a constant temperature and relative humidity under a regular light–dark schedule (12 h:12 h), fed with a normal rodent diet and with tap water ad *libitum*. Anemia rats were fed with a low-iron (5 mg/kg Fe^2+^, TD.81062, Teklad, Madison, WI, USA) diet (normal diet 35 mg/kg Fe^2+^), rodent chow for 27 weeks. In the aortic stenosis model, the animals underwent surgical ligation of the aorta. Cardiac function was monitored by consecutive echocardiography scans. On the final experiment day, the rats were sacrificed using anesthesia overdose. Blood samples for the complete blood count (CBC) and chemical profiles were withdrawn directly from the heart before it was finally stopped in diastole via potassium chloride (KCL) injection. The heart was excised and dissected, and the base was preserved for molecular analyses via flash freezing in liquid nitrogen and preserved at −80 °C until the time of the analyses. The cardiac apex was immersed in 4% paraformaldehyde and underwent paraffin embedding for histological analyses.

### 2.2. Echocardiography

For the echocardiographic measurements, the rats were lightly sedated using an intraperitoneal injection of 29 mg/kg ketamine and 4.3 mg/kg xylazine, placed in a left lateral decubitus position and scanned via a commercially available echo-scanner using a 10S phased array pediatric transducer and a cardiac application. The transmission frequency was 10 MHz, the depth was 2.5 cm, and the frame rate was 225 frames/s. The measurements included 2 parasternal short-axis sections at the papillary muscle (PM) and apical levels. To measure ventricular functions, the ejection fraction (EF) (stroke volume/LV diastolic volume) was determined using the Vivid i LV function software (version 9.1.0., GE, Haifa, Israel) and was estimated for each cardiac level separately.

### 2.3. Blood Analysis

Every month, under light anesthesia (29 mg/kg ketamine and 4.3 mg/kg xylazine), 1.5 mL of blood was withdrawn from the subclavian sinus of the anemic and AS rats. The blood was analyzed for CBC.

### 2.4. Euthanasia

On the last day of the physiological experiments, the rats were anesthetized using a high dose of anesthetic mixture (87 mg/kg ketamine and 13 mg/kg xylazine). The heart was stopped in diastole using an injection of KCl [1M] directly into the heart. The heart was excised, HW was established, and HW/BW was calculated. The heart was then dissected. The apex (AP) was preserved in 4% paraformaldehyde for histological analyses and the base was flash-frozen in liquid nitrogen and preserved at −80 °C for biochemical analyses.

### 2.5. Histology

The paraffin-embedded AP was sectioned into 5 μm sections using a microtome and was stained with hematoxylin and eosin (H&E). The sections were photographed using the Image J freeware (ImageJ 1.46r, National Institute of Health, Bethesda, MD, USA) and analyzed for structural measurements: cross-sectional area (CSA), LV-CSA, the LV cavity area, and the septal and anterior wall widths. 

#### 2.5.1. Light Microscopy

For the morphological assessment, histological sections were stained with hematoxylin and eosin (H&E) and were photographed and analyzed for the LV cavity area, LV muscle-CSA, and septal wall thickness.

#### 2.5.2. Scanning Electron Microscopy

Sections of each experimental group were scanned with an electron microscope for the characterization of the sarcomere structure. Five μm heart sections were placed onto glass cover slips (324 mm^2^). Sections were deparaffinized in an oven (65 °C for 30 min) and sequentially immersed in xylazine and 100% ethanol. Furthermore, they were treated with hexamethyldisilazane (HMDS) solvent twice. The HMDS was used to remove any liquid from the specimen. Furthermore, the samples were left to dry in a chemical hood for 10 min and were then sputter-coated with 2.5–3 nm iridium for 45 s on a Q150R S apparatus (Quorum) and imaged in a Merlin field emission scanning electron microscope (SEM) (Zeiss) (Fichman and Shaulov 2014). Using this method, the longitudinal and transversal sarcomere structures were characterized.

### 2.6. Protein Expression Analysis

Proteins were extracted from flash-frozen base cardiac tissues to produce whole-cell proteins (lysate). Using the Bradford method, the protein concentration was assessed, and 25 μg protein samples were loaded on 12% SDS polyacrylamide gel and separated by electrophoresis. Subsequently, the proteins were transferred to nitrocellulose membranes, and those membranes were then incubated for one hour in skimmed milk (1% fat) to block all non-specific sites. Subsequently, the membranes reacted with one of the following primary antibodies: ERK1/2, p-ERK1/2 or p-MEK1/2. All antibodies were obtained from Cell Signaling Biotechnology (Beverly, MA, USA). Each nitrocellulose membrane was then subjected to secondary antibody anti-rabbit IgG (Sigma-Aldrich, St. Louis, MO, USA). Finally, using an EZ-ECL substrate (Biological Industries, Beit Haemek Ltd., Beit Haemek, Israel) reaction, the membranes were scanned via G:BOX chemi XX6 (Lumitron, Petach-Tikva, Israel). The blots underwent densitometry using the ImageJ freeware (ImageJ 1.46r, National institute of health, USA) to quantify the expression of each protein. Protein expression was normalized to the expression of GAPDH (Sigma-Aldrich, St. Louis, MO, USA).

### 2.7. Statistical Analysis

Two-way analysis of variance (ANOVA) with repeated measures was used to evaluate differences in the repeatedly measured parameters, such as weekly blood hemoglobin and echocardiograph parameters, in which time (session) and treatment (low-iron diet/normal chow diet or AS/sham-operated) were the dependent variables. For the final Western blot analysis, as well as for the histological variables, one-way ANOVA was used. The Holm–Sidak method was used as a post hoc test whenever the ANOVA was significant. In the case of the two treatment groups, Student’s *t*-test was used. Values of *p* < 0.05 were considered statistically significant.

## 3. Results

### 3.1. Development of Anemia

Throughout 27 weeks of the experiment, rats in the low-iron diet group and rats in the control group gained weight similarly. Their initial body weight (BW) was ~280 ± 23 g., and at the end of the experiment it was ~500 ± 43 g. (Figure 1A). The hemoglobin level decreased in the low-iron diet group (starting in the seventh week), with a significant difference in the 12th week of the experiment and reaching a steady level thereafter (12 ± 3 g/dL vs. 15 ± 3 g/dL, *p* < 0.001, Figure 1B).

The anemic rats developed iron deficiency anemia with lower iron and higher transferrin levels (68.1 ± 22.6 µg/dL vs. 182.8 ± 70.7 µg/dL, *p* < 0.001; 170.5 ± 35.3 mg/dL vs. 120.2 ± 22.1 mg/dL, *p* < 0.01), respectively (Figure 2).

The red blood cell (RBC) concentration increased significantly in the anemic rats (*p* < 0.001) 4 and 5 months after low-iron diet initiation (Figure 3A). The mean corpuscular hemoglobin (MCH) and mean corpuscular hemoglobin concentration (MCHC) decreased significantly in the anemic rats (Figure 3B,C, respectively). Furthermore, the percentage of microcytic RBCs was increased (*p* < 0.001, Figure 3E); however, the mean corpuscular volume (MCV) and the percent of hypochromic cells were insignificantly changed (Figure 3D,F). Taken together, the hematological data indicate the development of significant microcytic hypochromic anemia consistent with iron deficiency.

### 3.2. Echocardiographic Parameters in the Two Models

We performed echocardiography at baseline and prior to sacrifice. Representative short-axis views of the diastole of normal and anemic rat hearts are depicted in Figure 4A,B, respectively.

The echocardiographic parameters obtained in the anemic and AS rats were consistent with eccentric and concentric hypertrophy, respectively. The anemic rats developed a larger LV diastolic diameter (LVDD) (7.2 ± 0.2 mm vs. 6.4 ± 0.3 mm, *p* < 0.001) with a thinner septal thickness (2.1 + 0.3 vs. 2.5 ± 0.2 mm, *p* < 0.05). In the AS model, a smaller LVDD (5.8 + 0.4 mm vs. 6.1 + 0.1 mm, *p* < 0.05) and thicker septum (2.8 + 0.6 vs. 2.4 + 0.1, *p* < 0.05) were observed. A representative M-mode echocardiographic image for LVDD calculation is provided in Figure 5. The LV ejection fraction, assessed at the end of the study, revealed preserved (or even, increased) contractility in the AS model (70.3% + 2.8 vs. 60.0% + 2.1, *p* < 0.05), while a decrease in LV function was observed in the anemic rats (52.6% + 4.7 vs. 60.3% + 2.8, *p* < 0.05).

### 3.3. Histological Results

In the histological sections, the LV-CSA was larger in the anemic rats (47.0 ± 8.1 mm^2^ vs. 59.9 ± 5.1 mm^2^, *p* = 0.002). The LV muscle CSA and total heart (left and right ventricles) muscle CSA were increased in the AS model (79.5 ± 9.33 mm^2^ vs. 62.2 ± 3.1 mm^2^ and 93.4 ± 13.8 mm^2^ vs. 73.8 ± 6.8 mm^2^, respectively, *p* < 0.001). Representative histological H&E sections of the anemic, AS, and control rats are shown in Figure 6.

Representative histological sections with H&E of the anemic, AS, and control rats are given in Figure 7.

The echocardiographic and quantitative histological parameters of the anemic and AS rats are summarized in Table 1 and Table 2, respectively.

We also examined the heart weight (HW) and the heart weight/body weight (HW/BW) ratio in the rats. In the anemic rats, we found no change in the HW (Figure 8A) or HW/BW ratio (Figure 8B).

In the AS model, both the HW and HW/BW ratios were higher compared to the control group (Figure 9).

### 3.4. Scanning Electron Microscopy

Differences in sarcomere structure in both the longitudinal and transverse positions are demonstrated via SEM in Figure 10 and Figure 11. The sarcomeres in the anemic group are thinner and circular (Figure 10), while in the AS specimen, the sarcomeres look elongated and thicker than in the normal heart section (Figure 11).

### 3.5. Biochemical Analysis

Our results show that in the anemic rats, the ERK1/2 decreased (*p* < 0.05), while the p-ERK 1/2 and the p-MEK 1/2 levels did not change significantly relative to the normalized protein GAPDH (Figure 12).

ERK 1/2 expression was increased in the AS rats compared to the control rats (*p* < 0.05) (Figure 13).

## 4. Discussion

We studied the relation between two different types of cardiac hypertrophy (concentric and eccentric) and ERK1/2 protein expression. The anemic rats developed an enlarged LVDD, thinner septum, and reduction in LVEF. The AS model rats had an increased HW/BW, thicker septum, and increased LVEF. The histological sections and SEM also supported the findings that aortic ligation induced AS, whereas a low-iron diet induced eccentric hypertrophy. During pressure overload, cardiac concentric hypertrophy aims to decrease wall stress and improve cardiac contractility. However, this compensatory mechanism may eventually fail due to increased oxygen demand and a reduction in blood supply to the hypertrophic myocardium, leading to irreversible changes in burned-out “cardiomyopathy” [19,20]. In eccentric cardiac hypertrophy (as seen in volume overload conditions and anemia), the compensatory LV dilation is also associated with elevation and may lead to cardiac failure [20]. In previous studies, Kehat et al. demonstrated that ERK1/2 knockout transgenic mice with a complete loss of ERK1/2 phosphorylation activity develop eccentric hypertrophy [21]. In our study, we showed, in a reverse pattern, that ERK 1/2 expression was influenced by the type of hypertrophy. However, it should be noted that we examined the models at two different stages, since the anemic rats had decreased LVEF reflecting the beginning of “burned out cardiomyopathy”. The interpolation of cardiac function makes the final relation between ERK expression and cardiac hypertrophy more complicated [22]. In one study on an AS model, the induction of early concentric hypertrophy followed by late eccentric hypertrophy (reflecting the transition from compensated to decompensated hypertrophy) was accompanied by subsequent alterations in ERK 1/2 expression [23]. On the cellular level, the sarcomeres became thinner and circular in the anemic model and thicker and elongated in the AS rats. These results are consistent with the published literature, suggesting a compensatory mechanism in which the recruitment of the sarcomeres in parallel or in series determines the type of hypertrophy as concentric or eccentric, respectively [24,25]. The alterations in ERK expression were influenced by structural changes and the time elapsed following the trigger onset. During concentric hypertrophy, ERK1/2 pathway activation contributes to increasing myocytes’ width and recruiting them in parallel, while in eccentric hypertrophy, the alteration in the pathway drives the cell toward cellular elongation. In accordance with our study, Toischer et al. compared two models of increased afterload and preload in mice via transverse aortic constriction and aortocaval shunt, respectively [26]. They showed that maladaptive cardiac hypertrophy in the afterload model was associated with apoptosis and fibrosis, whereas preload was associated with less apoptosis, better cardiac function, and no fibrosis [26]. Recently, Hartmann et al. also reported the results of in vitro mechanical overload on ejection heart models in rabbit and human isolated muscle strips and found that selective afterload elevation results in ERK phosphorylation, whereas preload activates the AKT-GSK3β pathway, which plays a role in preventing cell apoptosis [27,28].

### 4.1. Implications and Future Directions

The targeting of the ERK pathway to attenuate or reverse pathological hypertrophy has been studied in numerous models. Pimasertib, an MEK1/2 inhibitor, was shown to mitigate hypertrophic phenotype in mice with the activated MET tyrosine kinase receptor [29]. Other ERK inhibitors are also under pre-clinical evaluation in various types of cardiac hypertrophy [8].

### 4.2. Strength and Limitations

We provide further insight into the role of ERK1/2 in cardiac remodeling, since we examined models of common clinical scenarios using in vivo rat models and evaluated the hearts without any genetic manipulation. Presumably, the ERK1/2 pathway is the first to be affected by physical stress (pressure or volume overload) due to a yet-to-be-determined mechanism and the final impact is unaffected by its precursor, p-MEK1/2, or its activated form, i.e., p-ERK1/2. One of the limitations of our study is the difference in the stage of the remodeling process, as we evaluated the AS rats at the point when cardiac function was still preserved, while the anemic rats were in the decompensated stage of reduced LVEF. This difference may limit the interpretation of the results, since we could not isolate the relationship between hypertrophy and ERK expression.

## 5. Conclusions

ERK1/2 proteins play a central role in mediating adaptive and maladaptive cardiac hypertrophy. We showed, in a reverse pattern, that concentric cardiac hypertrophy upregulates ERK 1/2 expression, whereas eccentric hypertrophy induces its downregulation. This process is complex, and the final outcome depends, in part, on the type and duration of the stress and on cardiac function.

## Figures and Tables

**Figure 1 biomedicines-11-02337-f001:**
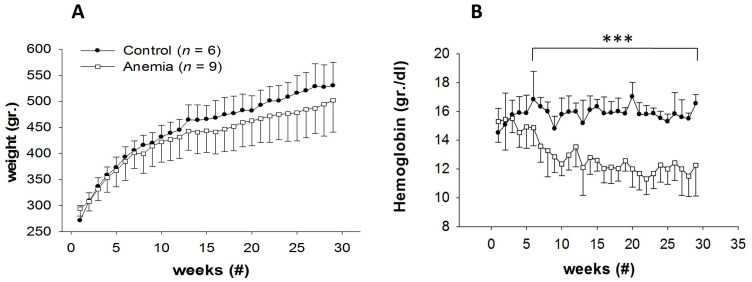
Body weight (**A**) and hemoglobin concentration (**B**) of the control (black circles) and low- iron-diet white circles) rats over 27 weeks of the experiment. *** *p* < 0.001 vs. anemia, (#) = number.

**Figure 2 biomedicines-11-02337-f002:**
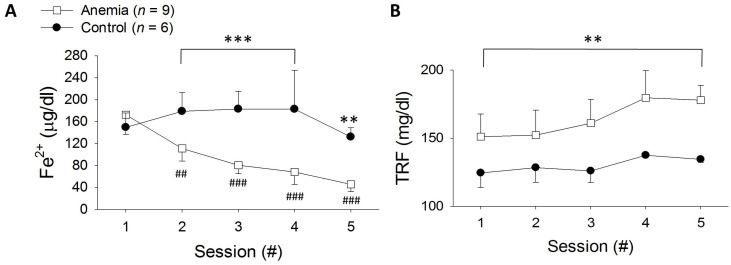
Iron (Fe^2+^) (**A**) and transferrin (TRF) (**B**) levels in the anemic and control groups. Control (black circles) and low-iron-diet rats (white squares) in each month of the experiment. ** *p* < 0.01 vs. anemia, *** *p* < 0.001 vs. anemia, ### *p* < 0.001 vs. baseline, ## *p* < 0.01 vs. Baseline, (#) = number.

**Figure 3 biomedicines-11-02337-f003:**
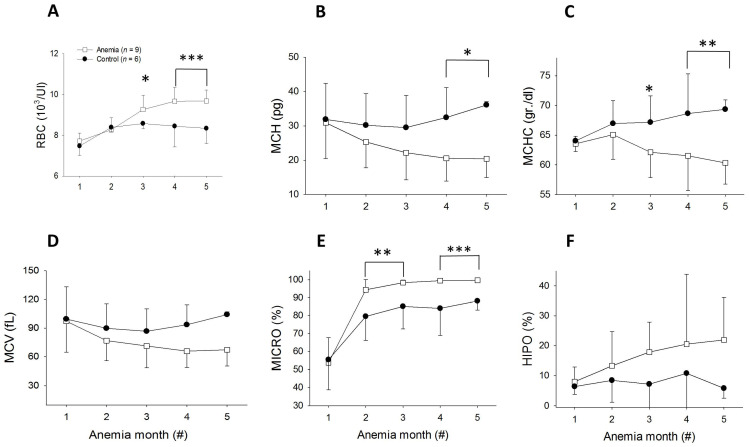
Complete blood count (CBC) results. RBC concentration (**A**), MCH (**B**), MCHC (**C**), MCV (**D**), percent of microcytic red blood cells (**E**), and percent of hypochromic red blood cells (**F**) in the control (black circles) and low-iron-diet rats (white squares) throughout the experiment. * *p* < 0.05 vs. control. ** *p* < 0.01 vs. control, *** *p* < 0.001 vs. control, (#) = number.

**Figure 4 biomedicines-11-02337-f004:**
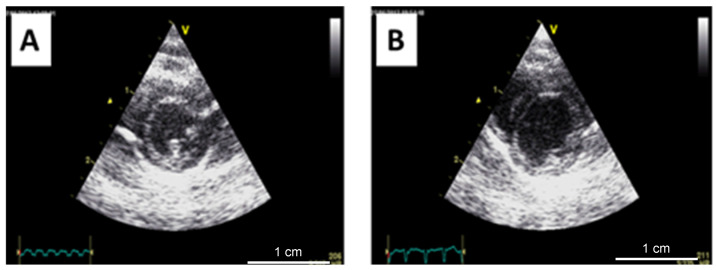
Representative echocardiographic images of control (**A**) and anemic rats (**B**) from the short-axis view at the papillary muscle level showing a dilated left ventricle in the anemic model.

**Figure 5 biomedicines-11-02337-f005:**
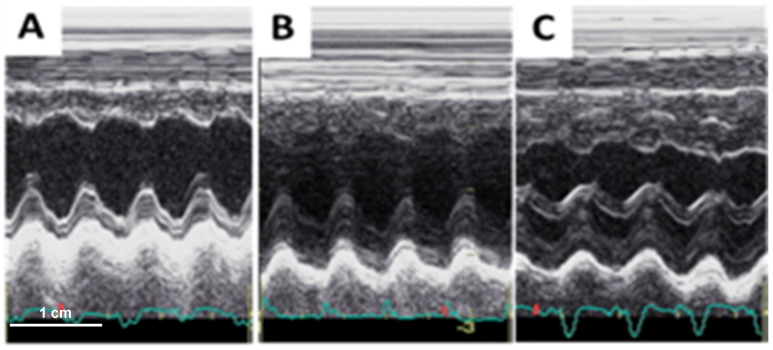
Representative echocardiographic images of anemic rats (**A**), aortic stenosis rats (**B**), and control rats (**C**). The LV diastolic diameter was measured in this view.

**Figure 6 biomedicines-11-02337-f006:**
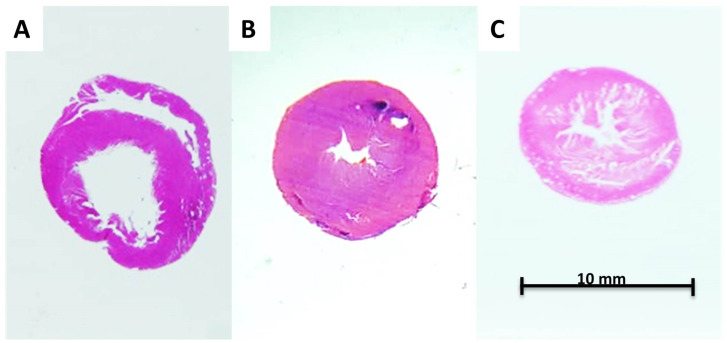
Representative histological (H&E-stained) sections of the hearts taken from the various models. (**A**) Anemia-eccentric hypertrophy heart. (**B**) AS-concentric hypertrophy heart and (**C**) control. The anemic heart cavity is wider than the control, and concomitantly, the AS sample is characterized by thicker LV walls. Magnification (×3).

**Figure 7 biomedicines-11-02337-f007:**
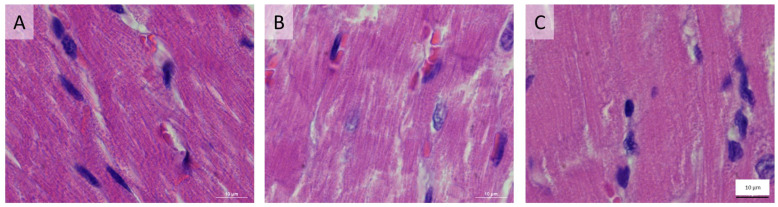
Representative H&E stain cross-sections from anemic (**A**), aortic stenosis (**B**), and control (**C**) groups. In the AS rats, the tissue is thicker and darker, while in the anemic rats, the tissue is sparse and thin compared to the control group. Magnification ×100.

**Figure 8 biomedicines-11-02337-f008:**
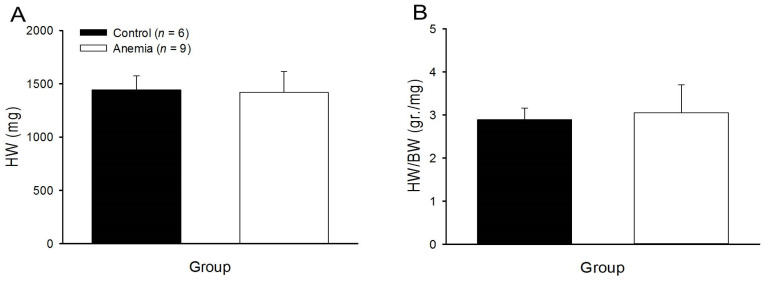
HW (**A**) and HW to BW ratio (**B**) of anemic (white column) and control rats (black column).

**Figure 9 biomedicines-11-02337-f009:**
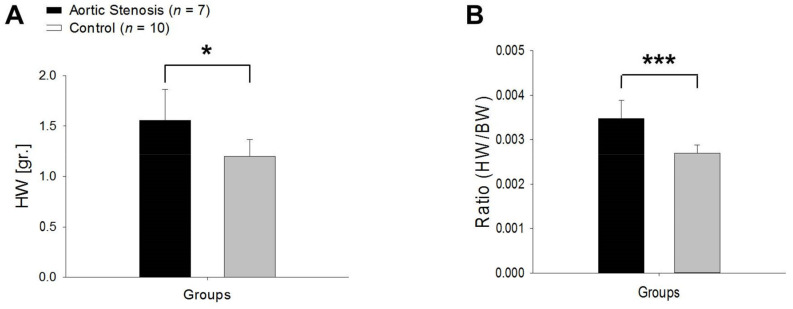
HW (**A**) and HW/BW ratio (**B**) in the AS rats (black column) vs. control rats (grey column). * *p* < 0.05 vs. control, *** *p* < 0.001 vs. control.

**Figure 10 biomedicines-11-02337-f010:**
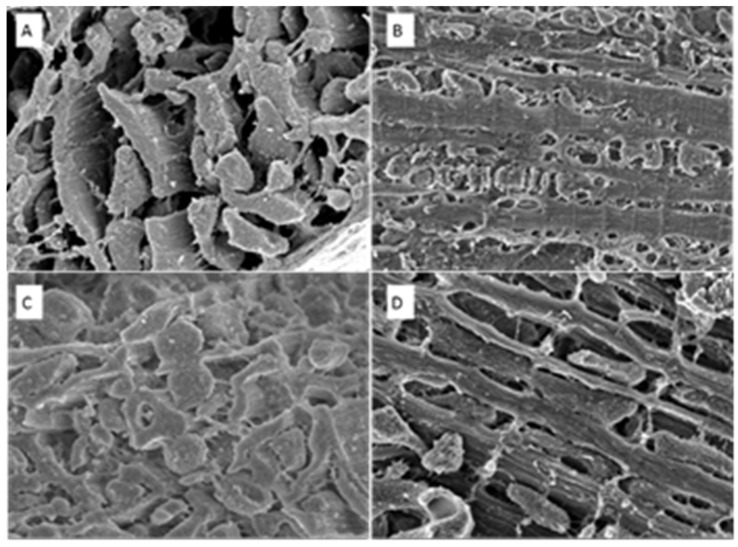
SEM of samples taken from the cardiac PM level. Transverse (**A**) and longitudinal (**B**) positions of control (normal) heart section. Transverse (**C**) and longitudinal (**D**) positions of anemic (eccentric) heart section. Magnification ×50,000.

**Figure 11 biomedicines-11-02337-f011:**
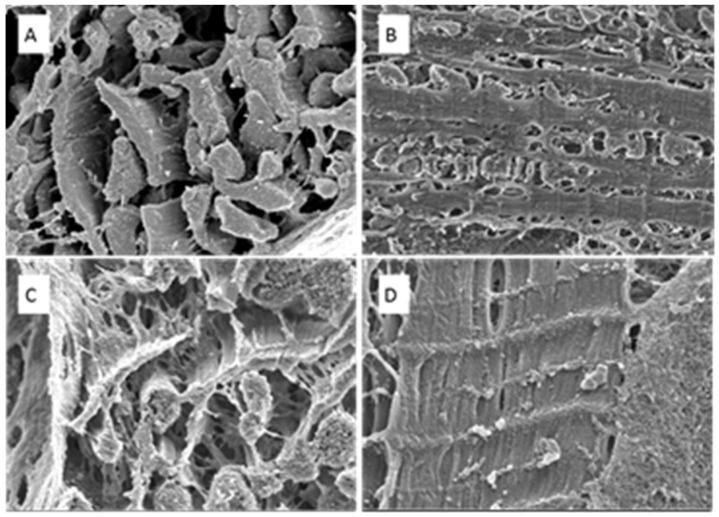
SEM images at the PM level. Transverse (**A**) and longitudinal (**B**) positions of control (normal) heart section. Transverse (**C**) and longitudinal (**D**) positions of AS (concentric) heart section. Magnification ×50,000.

**Figure 12 biomedicines-11-02337-f012:**
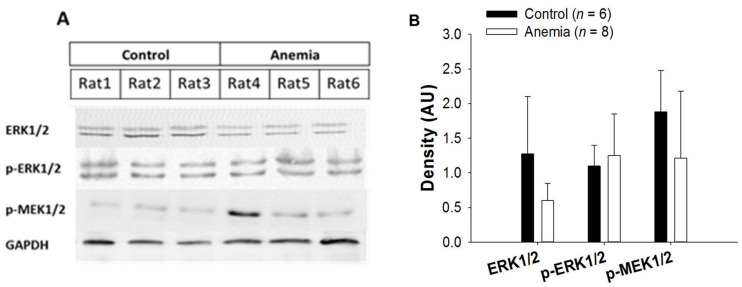
Representative Western blots of ERK1/2 (**A**) and densitometry analysis of Western blots (**B**) in anemic (white column), control (light grey column), and AS (black column) groups. Each column represents the mean ± SD density blots.

**Figure 13 biomedicines-11-02337-f013:**
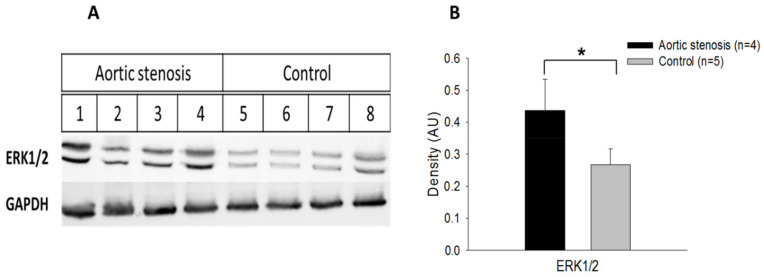
Representative Western blots of the indicated proteins extracted from the hearts of AS and control rats (**A**). Densitometry analysis (**B**) of Western blots. Control (black column) and AS rats (grey). Each column represents mean ± SD density. * *p* < 0.05 vs. control.

**Table 1 biomedicines-11-02337-t001:** Echocardiographic characteristics of the anemic rats.

	Anemia	Control	*p*-Value
*n*	9	9	
LVDD (mm)	7.2 ± 0.2	6.4 ± 0.3	<0.001
LV muscle CSA (mm^2^)	90 ± 13.2	75 ± 10.9	0.03
LV cavity CSA (mm^2^)	47 ± 8.1	59.9 ± 5.1	0.002
Septal thickness (mm)	2.1 ± 0.3	2.5 ± 0.2	<0.05
EF (%)	52.6 ± 4.7	60.3 ± 2.8	<0.05

CSA, cross-sectional area; EF, ejection fraction; LV, left ventricle; LVDD, left ventricle diastolic diameter.

**Table 2 biomedicines-11-02337-t002:** Echocardiographic parameters of the AS rats.

	AS	Control	*p*-Value
*n*	9	6	
LVDD (mm)	5.8 ± 0.4	<0.05	6.1 ± 0.1
Total cardiac muscle CSA (mm^2^)	93.4 ± 13.8	73.8 ± 6.8	<0.001
LV muscle CSA (mm^2^)	79.5 ± 9.33	57.9 ± 5.0	<0.001
Septal thickness (mm)	2.8 ± 0.6	2.4 ± 0.1	<0.05
HW/BW (mg/g)	3.48 ± 0.4	2.48 ± 0.8	<0.001
EF (%)	70.3 ± 2.8	60.0 ± 2.1	<0.05

AS, aortic stenosis; BW, body weight; CSA, cross-sectional area; EF, ejection fraction; HW, heart weight; LVDD, left ventricle diastolic diameter.

## Data Availability

Not applicable.

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
