# Peer review of "The Role of Extracellular Signal-Regulated Kinase Pathways in Different Models of Cardiac Hypertrophy in Rats"

_biomedicines, 2023, doi:10.3390/biomedicines11092337_

Round 1
Reviewer 1 Report
The study is well conducted and written. Some changes are needed:
Â
The abstract appears poor, it can enriched with some additional statements
The introduction is broad need to be shortened and much more focused
The discussion section should be re-structured
·     Main findings and comparisons with similar previous ones.
·     Potential implications
·     Limitations and strength
·     New direction for future research
Â
Figure 5 is of poor quality need to be improved
Author Response
Thank you for considering our manuscript for publication. We also wish to thank the reviewers for taking the time to review the paper and for their helpful comments.
Reviewer 1
.The abstract appears poor, it can enriched with some additional statements.
The abstract has been revised.
.The introduction is broad need to be shortened and much more focused.
The introduction has been revised and shortened.
.The discussion section should be re-structured
     The discussion has been re-structured with the proposed sections
Figure 5 is of poor quality need to be improved.
Another Figure is given, hopefully with higher resolution.
Reviewer 2 Report
This paper explored the relationship between cardiac hypertrophy and the ERK ½ pathway in rats. The authors induced eccentric hypertrophy through a low-iron diet and concentric hypertrophy through transverse aortic constriction in a rat model. In the eccentric model, they noted a reduction in ERK ½ expression, coupled with left ventricle cavity dilation, ventricular wall thinning, and diminished left ventricle function. Conversely, the concentric model exhibited elevated ERK ½ expression, thicker ventricular walls, and enhanced heart contractility. The study is well-designed, yielding clear results. However, the following concerns need to be addressed.
1.    To enhance readability, it is recommended to rephrasing sentences that heavily rely on "respectively." This will help prevent potential confusion among readers who are absorbing substantial information.
2.    The authors can consider utilizing scatter plots for graph representation, as this visualization method offers a clearer depiction of individual values. This will enable readers to discern each data point with greater precision.
3.    It is helpful if the authors expand on the significance of the findings. Provide a more comprehensive discussion of how these findings contribute to the broader context of cardiac hypertrophy research. Elaborating on the potential implications for clinical interventions or therapeutic strategies could further underscore the importance of the study's outcomes.
Author Response
Thank you for considering our manuscript for publication. We also wish to thank the reviewers for taking the time to review the paper and for their helpful comments.
Â
Reviewer 2
- To enhance readability, it is recommended to rephrasing sentences that heavily rely on "respectively." This will help prevent potential confusion among readers who are absorbing substantial information.
Â
         The whole manuscript has been revised.
Â
- The authors can consider utilizing scatter plots for graph representation, as this visualization method offers a clearer depiction of individual values. This will enable readers to discern each data point with greater precision.
Unfortunately, we couldn't apply this for all our figures since only Figures 1-3 may be depicted with such representation. All the remaining figures are mostly histological.
Â
- It is helpful if the authors expand on the significance of the findings. Provide a  more comprehensive discussion of how these findings contribute to the broader context of cardiac hypertrophy research. Elaborating on the potential implications for clinical interventions or therapeutic strategies could further underscore the importance of the study's outcomes.
Â
The discussion section has been revised focusing on the following points:
Main findings, comparison with previous studies, clinical implications and future possible directions, and limitations and strengths.
Â
Â

Round 2
Reviewer 1 Report
NonÂ